# STING Agonists/Antagonists: Their Potential as Therapeutics and Future Developments

**DOI:** 10.3390/cells11071159

**Published:** 2022-03-29

**Authors:** Danilo Guerini

**Affiliations:** Novartis NIBR ATI, 4056 Basel, Switzerland; danilo.guerini@novartis.com; Tel.: +41-79-292-58-70

**Keywords:** cGAS, STING, drug discovery

## Abstract

The cGAS STING pathway has received much attention in recent years, and it has been recognized as an important component of the innate immune response. Since the discovery of STING and that of cGAS, many observations based on preclinical models suggest that the faulty regulation of this pathway is involved in many type I IFN autoinflammatory disorders. Evidence has been accumulating that cGAS/STING might play an important role in pathologies beyond classical immune diseases, as in, for example, cardiac failure. Human genetic mutations that result in the activation of STING or that affect the activity of cGAS have been demonstrated as the drivers of rare interferonopathies affecting young children and young adults. Nevertheless, no data is available in the clinics demonstrating the therapeutic benefit in modulating the cGAS/STING pathway. This is due to the lack of STING/cGAS-specific low molecular weight modulators that would be qualified for clinical exploration. The early hopes to learn from STING agonists, which have reached the clinics in recent years for selected oncology indications, have not yet materialized since the initial trials are progressing very slowly. In addition, transforming STING agonists into potent selective antagonists has turned out to be more challenging than expected. Nevertheless, there has been progress in identifying novel low molecular weight compounds, in some cases with unexpected mode of action, that might soon move to clinical trials. This study gives an overview of some of the potential indications that might profit from modulation of the cGAS/STING pathway and a short overview of the efforts in identifying STING modulators (agonists and antagonists) suitable for clinical research and describing their potential as a “drug”.

## 1. The cGAS/STING, a Nucleic Acid Sensing Pathway, Plays a Role in Many Diseases

Nucleic acids are an important component of the cell. They store genetic information and provide guidance to the cell on how to execute it. Nevertheless, when nucleic acids are found outside the cell or when large amounts of them are misplaced in the cytosol, which occurs because of damage to the cell (intrinsic cell death, viral infection, mitochondria damage), nucleic acids are recognized as harmful agents (as pathogen-associated molecular patterns or “PAMPs”) and trigger a strong immunological response. Such a response is observed in many autoinflammatory and autoimmune diseases, where the activation of nucleic acid sensors has been suggested as a major molecular determinant driving the pathology [1].

Two novel gene products (cGAS and STING) have been recently identified as the key players in the recognition of excess cytosolic dsDNA [2,3,4]. Upon binding to dsDNA, cGAS (a cyclic GMP/AMP synthase) converts GTP and ATP to the cyclic dinucleotide called cGAMP [3]. STING (Stimulator of Interferon Genes) [5] binds cGAMP, undergoing a conformational change, thereby facilitating the phosphorylation of the transcription factor IRF3, finally leading to a large increase in the expression of type I IFN genes [6]. cGAMP is a cyclic dinucleotide composed of one molecule of GMP and one of AMP, with a very unusual 2′,5′ linkage and a classical 3′,5′ linkage [7], and it represents a novel “2nd” messenger. Activation of this pathway leads to a strong type I interferon response which is generally paralleled by an increase of transcription of many ISG (interferon-stimulated genes). Diseases showing a strong type I IFN signature are defined as interferonopathies [8].

A well-characterized genetic-linked interferonopathy is the so-called Aicardi-Goutières-Syndrome (AGS). In around 25% of AGS patients, uncontrolled type I IFN response is linked to mutations of the cytosolic DNase Trex1, which results in an increase of cytosolic dsDNA that activates cGAS. A similar mechanism is common to AGS patients, who have mutations in other DNA processing enzymes (RNASEH2A, RNASEH2B, RNASEH2C, and SAMHD1) [9]. The clinical manifestations in AGS patients are very similar to those observed in lupus patients. A milder form of the disease is found in Familial Chilblain Lupus patients, who are carrying a heterozygous mutation in Trex1 [10]. Among the Mendelian diseases related to TREX1 loss-of-function mutation, a less severe form leads to RVCL (autosomal dominant retinal vasculopathy with cerebral leukodystrophy), which is characterized by an adult-onset of vasculopathy, leading to retinopathy and juvenile ischemic stroke [11]. STING-associated vasculopathy with onset in infancy (SAVI) is another lupus-like disease with a link to the cGAS/STING pathway that is the consequence of the uncontrolled activation of the pathway. Identified as one of the interferonopathies observed prevalently in young persons, this disease was shown to be the consequence of mutations hyperactivating STING, resulting in a chronic type I IFN response. Manifestations of this pathology are evidenced by skin rashes, lung inflammation, and inflammation in the extremities, leading in extreme cases to amputation [12]. Diseases with a defect in the DNA-processing enzymes (as for the Trex AGS patients) are expected to respond well to cGAS inhibition: preclinical work has documented that the increase of cytosolic dsDNA leads predominantly to the activation of cGAS, while the contribution of other DNA sensors such as AIM2 seems to be minor [13]. In contrast, for SAVI patients, STING antagonists are the therapeutics of choice, although it is not yet known if one compound might be capable of blocking, to the same extent, all (hyperactivated) STING mutants that have been identified so far. 

Learning from AGS and SAVI patients might teach us where, in man, the cGAS/STING pathway plays an important role: for example, prominent damage of blood vessels has been observed in SAVI patients [12], suggesting that activation of STING might play a role in some of the non-genetically linked vasculitis disorders, although evidence for the latter is still fragmentary.

Besides the rare genetic diseases, there is evidence suggesting that the cGAS/STING pathway might play a role in chronic diseases, where programmed cell death is not efficiently clearing cellular debris [14]. In lupus patients, the chronic damage of different organs leads to the appearance of antiDNA antibodies, which suggest a contribution of the cGAS/STING pathway in this disease [15].

Diseases such as subtypes of systemic lupus erythematosus (SLE), lupus nephritis (LN), and dermatomyositis, which have been suggested to be triggered by DNA viruses such as EBV, cytosolic dsDNA, or mitochondrial dsDNA, are also expected to be driven (at least in part) by the aberrant activation of cGAS. A prominent role of cGAS in the development of Sjogren’s Syndrome (SS) would not be unexpected since this disease shares many similarities with SLE; this has been confirmed by a recent publication showing hypersensitivity to cGAS/STING activators in SS patients [16]. 

Low molecular weight inhibitors of cGAS might also be effective in treating the skin rashes/reddening associated with SLE, a pathology that is often observed when SLE patients are exposed to UV light [17]. There has also been some evidence that deposition of dsDNA in joints might be responsible for the inflammation observed in rheumatoid arthritis patients [18], although it is unlikely that cGAS/STING inhibitors will be superior to the TNF blockers that are used in the clinics: evidence from preclinical models suggests that TNF also controls, among others, the cGAS activation in joint inflammation [19].

Some reports have suggested cGAS/STING modulation as a potential treatment of ulcerative colitis and inflammatory bowel disease (IBD) [20,21,22,23]. At the same time, other reports showed that blocking the cGAS/STING pathway worsened the outcome of colitis [24]. In the case of these diseases affecting the gastrointestinal tract, we need to better understand the role of the microbiome in modulating the cGAS/STING pathway since evidence has accumulated that the bacteria producing cyclic dinucleotides are capable of activating STING [25]. Based on recent observations, it was proposed that blocking cGAS/STING might show some efficacy in inflammation driven by sepsis [26], although to achieve a robust clinical remission in this disease, it might require combining cGAS/STING inhibitors with drugs targeted to TLRs and other DNA sensing pathways.

A large body of evidence has indicated that both cGAS and STING are involved in lung inflammation. Damage to lung epithelial (by different agents) causes the release of DNA, appearing in bronchoalveolar lavage (BAL), which seems to be sufficient to activate the cGAS/STING pathway. Intratracheal application of DNAse strongly reduced the type I IFN response in a model of silicosis-driven lung inflammation, suggesting that the released DNA has an inflammatory effect [27]. Different publications using genetically modified animals confirmed the role of STING in models of lung inflammation [27,28,29]. Although therapeutic intervention in the cGAS/STING pathway might lead to some improvements in diseases such as cirrhosis and endomyocardial fibrosis [30,31,32,33,34,35], more data would be needed to confirm a strong general effect in fibrosis. Aberrant cGAS/STING activation might also play a role in diseases such as nonalcoholic steatohepatitis (NASH) and chronic obstructive pulmonary disease (COPD); nevertheless, this still requires further evaluation. In a mouse model of SAVI patients, conditions were identified under which bacterial-derived cyclic nucleotides were driving lung inflammation. However, the same mice, which were made germ free, uncovered an unexpected protective function of the cGAS/STING pathway [36]. This suggests that contributions by the cGAS/STING pathway could, at the same time, worsen or protect disease pathology: it will be critical to figure out which of the two components is having the strongest effect before moving to a clinical setting.

cGAS and STING have been shown to play a role in cellular senescence [37,38], and there is some evidence that such findings will have an impact beyond the cellular phenotype [39]. To the protective effect, it needs to be considered that enhancing the survival of cells might lead to an increased risk of tumorgenicity. There is accumulating evidence that cGAS activation is involved in neuroinflammatory diseases such as Parkinson’s disease (or at least a subtype of them) [40], Alzheimer’s disease [41], and amyotrophic lateral sclerosis (ALS) [42]. In these latter diseases, controlling the cellular senescence component might have a therapeutic advantage. 

Both inhibiting cGAS and STING promoted recovery from acute kidney injury induced by cisplatin [43]. Novel data suggest that inhibiting the cGAS/STING pathway might also be beneficial in treating ApoL1 nephropathy [44].

Although the cGAS/STING pathway activation is considered one of the first defenses that the immune system deploys to fight against viral infection, once the acute phase is terminated, elevated type I interferon has been shown to propagate chronic inflammation that damages tissue and prevents tissue recovery [45,46]. It is, therefore, tempting to speculate that blocking the cGAS/STING pathway post-acute phase will accelerate the recovery from chronic viral damage.

## 2. How to “Control” the cGAS STING Pathway

There are different sites (targets) at which a low molecular weight inhibitor could modulate the cGAS STING pathway (Figure 1): cGAS, Trex1 DNase or other DNases, STING, ENPP1 (the enzyme degrading cGAMP), and cGAMP transporters. The inhibition of the cGAMP-degrading enzyme ENPP1 would result in the increase of the level of intracellular cGAMP and would keep the pathway activated [47].

This strategy is very similar to activating STING or activating cGAS by blocking DNA degradation. It might also be possible to prevent the transport of cGAMP via some of the recently identified transporters [48,49,50,51,52]: this might prevent the spreading of activation of the pathway to adjacent cells [53] or else maintain a high level of cGAMP at the primary site of its synthesis. Preventing cGAMP transport might overlap, in some cases, with the effect of inhibition of STING or cGAS (Figure 1). There have been large efforts to develop STING low molecular compounds, which is, in fact, the “only” site in this pathway for which both antagonists and agonists have been identified. 

## 3. STING Agonists in Cancer

5,6-dimethylxanthenone-4-acetic acid (DMXAA) had been, for a long time, considered a promising tumor drug candidate; it belongs to a class of compounds with strong antivascular activity. In view of the very promising rodent preclinical l data, this compound was moved in human clinical trials [54]. DMXAA and related compounds, despite their acceptable pharmacokinetic properties, failed to show any effect in human patients. It was demonstrated that DMXAA and other flavonoids were highly specific mouse STING agonists [55], which provided a first glimpse at the mode of action of this class of drugs. At the same time, data showed that DMXAA was completely inactive at human STING, explaining the lack of efficacy of this drug in human clinical trials [55]. Great effort was made to chemically modify DMXAA to gain activity at human STING. Despite very elegant early crystallography studies [56], which gave insight into the DMXAA-STING structure, it was not possible to find a derivative with good activity at human STING. Activities in this direction have now been paused since interesting novel STING agonists have been identified (Table 1). 

The lessons learned from mouse biology and the realization that the activation of STING leads to a strong killing of tumor cells, mediated by type 1 IFN response, have, in between, gained the interest of the oncology community. After the identification of cGAMP as the product of the dsDNA-mediated activation of cGAS, it was shown that this molecule had tumor suppressive activity in different preclinical tumor models. Many pharmaceutical companies and academic institutions launched chemical optimization efforts to transform cGAMP into a “drug-like” molecule. The investments seemed to pay off, as shown by the good efficacy of ADU-S100 in a mouse tumor model [57]. ADU-S100 had comparable activity toward mouse and human STING, making it the first candidate to move to early clinical studies. One of the challenges encountered with the cGAMP derivatives was how to circumvent their strong systemic effect upon oral/intravenous application. Therefore, early clinical trials were started with the local application of ADU-S100 at the tumor site. These trials did not progress as quickly as hoped, and some were terminated ahead of time [58]. The final reports on the outcome of these studies have not been fully published, but it is fair to assume that the efficacy of this STING agonist, as a single therapeutic agent, was not sufficiently convincing to progress it further in the clinics. Combinations of STING agonists with other cancer drugs is still an appealing strategy, although it has been difficult identifying the ideal pathway(s)/drug for such combination trials. These studies also underline how difficult it is to move from very successful animals model studies to human patients. 

There are many excellent summaries of the large amount of data that has become available about the experience with STING agonists in the clinics and beyond; these data will not be discussed in this review [58,59,60,61,62,63,64].

Novel STING agonists have recently been identified. These molecules are not derivatives of cGAMP, and they have been generated starting from different chemical spaces. These molecules should allow the generation of drugs with much better pharmacokinetic properties. Ramanjulu and colleagues at GlaxoSmithKline [65] succeeded in producing very potent amidobenzimidazole STING agonists with promising in vivo activity. Even more exciting, the amidobenzimidazole derivatives were recently shown to be efficacious in preventing viral spread in models of SARS-CoV-2 virus infection after intranasal application [66,67,68]. These data show a very elegant way of taking advantage of the strong effect of STING agonists in the respiratory tract while preventing, at the same time, the systemic exposure that would ensue from oral dosing, which might limit their application. The induction of a strong type I inflammation in the respiratory tract might be a strategy to fight inflammation driven by any viruses entering via the respiratory tract.

A STING agonist, showing pH-dependent dimerization, has been described [69]. While the monomer has weak activity, upon dimerization, the compound shows a strong STING agonistic effect. In in vivo preclinical experiments, pH-dependent dimerization occurred preferentially at the tumor site, while the monomer was observed at high level only in circulation, resulting in a low systemic effect upon oral delivery. Although it is early times, this might be a very appealing strategy to reduce side effects of systemic exposure of STING agonists.

## 4. STING Antagonists in Inflammatory Diseases

While STING agonists are expected to be of value mostly for cancer therapy, STING antagonists have a chance to find their home in many diseases that have a strong innate immune component. As summarized above, based on a large set of preclinical observations, there are several indications that might profit from therapeutics capable of tuning down the cGAS/STING pathway. After the early unsuccessful attempts to generate inhibitors starting from STING agonists, efforts were focused on trying to find completely novel chemical starting points. Compound screenings based on different strategies reported initial weak hits, most of them showing limited selectivity and lacking in vivo activity. It was, therefore, surprising when Haag and colleagues [70] reported the identification in a medium-throughput screen of small covalent inhibitors showing highly specific effects on STING. Up until then, the strategy for identifying STING inhibitors was based on searching for compounds that would prevent cGAMP binding to STING. Nevertheless, unbiased cellular screens, as carried out by Haag and colleagues [70], showed that with a well-designed approach, it was possible to identify STING antagonists with unexpected properties. The covalent inhibitors, called C-176 and H-151, bind in the region that connects the transmembrane domain with a large cellular portion of STING, thereby preventing STING from acquiring an “active” conformation. These inhibitors showed robust activity in vivo, and they were capable of reversing strong tissue inflammation associated with the chronic activation of the cGAS/STING pathway in Trex KO mice. These mice have been shown to recapitulate some of the disease characteristics of AGS patients [71,72,73]. These are very early molecules, and translating these covalent compounds to therapeutics will require some effort. These compounds have, nevertheless, generated significant interest from pharmaceutical companies, and it is expected that joint efforts will soon identify candidates for early clinical exploration. Just recently, Hong and colleagues [74] identified a class of STING inhibitors capable of competing for the binding of cGAMP to STING. The effort, led by Chinese academic institutions, started from in silico modeling, complemented by the screening of chemical libraries for binders to the soluble cyclic-dinucleotide binding domain of STING. One of these compounds, named SN-11, showed activity in vivo, with comparable efficacy to the covalent STING antagonist C-176. As for C-176, SN-11 showed good efficacy in the Trex1 KO model [74].

**Table 1 cells-11-01159-t001:** In vivo active STING agonists/antagonists with the potential as therapeutics.

STING Agonist	Publication	Characteristic	Mode of Action
Amidobenz-imidazole (diABZI) 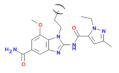	[65]	An agonist with nM affinity to STING once it forms a dimer (diABZ). The monomer has weak activity to STING: A monomer summarizing the basic chemical properties of the amidobenzimidazole is depicted	diABZI bind in the C-terminal domain of STING in the open conformation, e.g., like-cGAMP. The compound shows activity in preclinical tumor models. Recently, a derivative of diABZI showed efficacy in an animal model for SARS-CoV-2 virus infection [66]
MSA-2 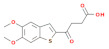	[69]	The monomer of the compound called MSA-2 forms a pH-dependent dimer that shows high affinity to STING. (20–100× fold increased affinity compared to monomer).	The MSA-2 is a weak acid, and it is preferentially taken up in an acidic tumor environment, where it can form a dimer, showing low systemic effects.
**STING Antagonist**	**Publication**	**Characteristic**	**Mode of Action**
H-151 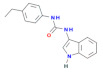	[70]	H-151 is a small covalent inhibitor of STING, depicted shows both mouse and human activity. Another class of small covalent inhibitors (C-176), with a different structure from H-151, shows a preferential effect at mouse STING	The compound H-151 binds to the stalk region of STING, preventing the dimerization (multimerization) required for the activation of STING. Both H-151 and C-176 have good in vivo activity, showing positive effects in the Trex1 KO mouse model
SN-11 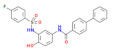	[74]	SN-011 is a novel STING inhibitor that targets the cyclic dinucleotide binding pocket and shows good efficacy in vivo. It has been suggested to prevent cGAMP binding to STING and, therefore, prevent its activation	SN-011 depicted in the left panel has been shown to work in vivo as a potent inhibitor of the cGAS/STING pathway. The SN-011 shows similar efficacy as the covalent compound H-151 in the Trex1 KO mouse model

## 5. Conclusions

The cGAS/STING pathway has become one of the currently most studied innate pathways, having been shown to play an important role in many preclinical models of inflammatory and chronic diseases. It is also very exciting that novel data suggest that modulating this pathway might provide us with new medications for fighting viruses affecting the respiratory tract or for addressing neuroinflammatory diseases. To further understand the full potential of this pathway in the clinics, it will be necessary that safe, potent compounds (both agonist and antagonist) become available to the clinicians in the next few years in order to start exploring the modulation of STING in human diseases. cGAS inhibitors (not discussed in this review) seem to be slowly catching up with STING agonist/antagonists. We might, therefore, soon have at hand a set of new compounds that act at different sites of this pathway, which should allow us to develop safe and efficacious therapeutics for a wide range of diseases.

## Figures and Tables

**Figure 1 cells-11-01159-f001:**
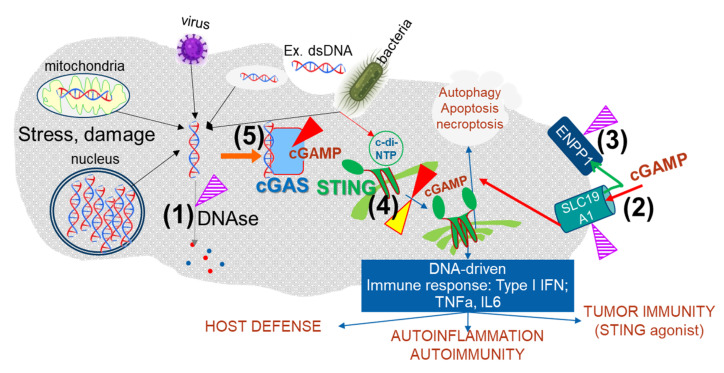
An increase in dsDNA in the cytosol can be the result of bacterial or viral infections. The entry of extracellular (ex.) DNA via endosomes results in the escape of partially digested DNA to the tightly controlled different DNAses, whose expression might vary from cell to cell. Bacteria might also activate the pathway by bypassing cGAS since cyclic dinucleotides produced by bacteria have been shown to bind to STING: cGAS (blue) is activated when cytosolic DNA is increased, and it synthetizes cGAMP. After binding cGAMP (red), STING dimerizes/multimerizes (green) and promotes the transcription of many cytokines belonging to the IFN type I family. The DNA-driven immune response is responsible for tumor immunity and plays a pivotal role in autoinflammation and autoimmunity. The cGAS/STING pathway can be modulated at different sites. Inhibition of the DNAses (1), the cGAMP transporters (shown using the example of SLC19A1) (2), or the cGAMP degrading enzymes ENPP1 (3) results in the increased activity of the pathway. Based on the current experience in drug discovery, it is unlikely that we will ever find low molecular weight compounds capable of stimulating at (1), (2), and (3); therefore, “inhibition” (violet-striped triangles) is to be considered the only therapeutic option. Using STING agonists (yellow triangle) (4) is the other option considered for activating the pathway. The activation of the cGAS/STING pathway has been shown to have a large potential for fighting tumors but might also be valuable in cases where a strong transient increase of the IFN response could help fight viral infection. Inhibition of the cGAS/STING pathway can be achieved with cGAS (5) or STING (4) specific inhibitors (red triangle). This intervention might be relevant for many autoinflammatory diseases that show an increase in IFN type I response. Evidence suggests that the cGAS/STING pathway controls autophagy and it has a role in apoptosis/necrosis. The relevance of these branches of the cGAS/STING pathway for their potential role in disease pathology is currently not well understood.

## Data Availability

Not applicable.

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
