# Peer review of "STING Agonists/Antagonists: Their Potential as Therapeutics and Future Developments"

_cells, 2022, doi:10.3390/cells11071159_

Round 1

Reviewer 1 Report

This review paper analyses the cGAS STING cellular pathway and its relationship with human diseases. The topic is interesting and the manuscript is well written. According to my opinion before considered for publication some parts have only to be reorganised providing a more detailed and focused conclusion, highlighting the main findings.

Detailed comments

lines 49-52: These mutations were statistically significant higher in AGS patients? If so, it is better to be stated here so that they could be associated with this syndrome

In the first section the authors discuss the diseases in which cGAS/STING may be or is involved. Since the study is a review paper it could be informative to present these diseases in a Table, possibly with an evaluation of how much evidence is available and proven for this.

line 144: add “of” after “increase”

The part in lines 174-176 sounds like the scope, so it is better to rephrase this statement or move it in the end of the first section. 

The second column in Table 1 should only have the number of the reference, not the title of the paper. This is given in reference list

Author Response

I would like to thank the reviewer with the many excellent suggestions, which I definitively found have helped the improve the quality of the manuscript. Details of the changed in the revised manuscript are listed below. I just wanted to discuss the suggestion of presenting the diseases where cGAS/STING might be involved in a Table. I have attempt to do so, but after a first draft of the Table, I was not sure on how to avoid many "repetitions" with the corresponding text, in particular if the evidence for each disease should also be presented in the table. A solution would be to remove most of the text discussion the diseases (lanes 38-142) and focus mainly on a table, which will be filled by quite some details. What would be the preference by this reviewer? In addition, I would also need to have the confirmation from the editorial office that an additional table would be acceptable.

Specific comments:

Lanes 49-53:  The paragraph has been edited and the information requested have been included. Hopefully the discussion on the mutations is now clearer.

Lane 144, now lane 147, “of” has been inserted

Lanes 174-176 (old) new 177-179 has new been changes as suggested by the reviewer

Table I has now re-edited an as suggested only the reference number has been included

Reviewer 2 Report

GENERAL COMMENTS

The review aims to discuss recent advances in cGAS/STING involvement in pathogenic conditions along with a comment on the discoveries of small molecules targeting the key immune modulator STING, highlighting both academic and industrial efforts to push those compounds into clinics.

While the concise nature of the review is admirable and the paragraphs provide an adequate glimpse into the topic, some aspects would benefit from a deeper discussion, or at least, I would suggest adding some important citations to guide the readers and provide connections and further readings to understand the complexity of the topic.

I noted there are several recent reviews describing in detail the on-going discovery and clinical developments of STING modulators, as well as their potential applications in cancer treatments or inflammatory disorders. Since the focus of this review is not to provide a list of clinical trials targeting STING nor a complete resource of STING modulators, I would suggest mentioning at some of the recent reviews covering the topic.

(i.e. Decout et al. 2021 - https://doi.org/10.1038/s41577-021-00524-z, Amozegar et al. 2021 - https://dx.doi.org/10.3390%2Fcancers13112695, Motedayen Aval et al. 2020 - https://dx.doi.org/10.3390%2Fjcm9103323, Ding et al. 2020 - https://dx.doi.org/10.1016%2Fj.apsb.2020.03.001, Hong et al. 2022 - https://doi.org/10.1093/jmcb/mjac005, Li et al. 2021 - https://dx.doi.org/10.3389%2Ffphar.2021.779425, Liu et al. 2021 - https://doi.org/10.1016/j.ejmech.2020.113113).

Recently, beside small molecules, also engineered viral peptides have been exploited to modulate STING and dampen IFN-I in cells derived from lupus patients (Prabakaran et al. 2021, EBioMedicine - https://dx.doi.org/10.1016%2Fj.ebiom.2021.103314): Could this be worth of mention in the review as a potential developmental strategy to target STING?

Below here I listed some minor corrections and suggestions that could help improving the review:

Line 22 – Why the author used “we” instead of “I”?

Line 40 and 44 – the author refers to cGAMP as a cyclic nucleotide, the correct nomenclature would be cyclic dinucleotide.

Line 48 – This sentence introduces a long and complex chapter, in which only some examples of Interferonopathies have been highlighted. It could be good to add at this point a more exhaustive review to refer to. For instance: Crow and Stetson, 2021. Nat. Rev. Immunol. https://doi.org/10.1038/s41577-021-00633-9

Line 56 – Citation for RVCL is missing?

Line 68 – This statement could be accompanied by citing the seminal work saying that mouse AIM2 and human IFI16 are dispensable for IFN-I production, whereas cGAS is not (Gray et al. 2016 – Immunity. https://doi.org/10.1016/j.immuni.2016.06.015)

Line 70 – Replace “extend” with “extent”

Line 72-74 Discuss if vasculitis directly depends on elevated type-I IFN, or on STING via different pathways.

Line 86 – Since Sjogren syndrome is cited and a potential involvement of cGAS is speculated, the author could insert a citation to the recent report of Huijser et al. 2022. Rheumatology - https://doi.org/10.1093/rheumatology/keac016, which actually suggests a trend towards hyperresponsiveness to STING ligands in primary Sjögren’s syndrome. At the end of the paragraph it may worth mentioning that also monogenic primary immune-deficiencies revealed unexpected link to exhacerbated IFN-I responses via cGAS-STING activation (10.1172/jci.insight.132857 )

Line 93 – What’s “Tonfa”?

Line 100 – Replace “It is was” with “It was”

Line 112 – It is not clear and it could be probably expanded the comment on idiopathic pulmonary fibrosis (IPF). What does the author mean by saying that the role of cGAS/STING in IPF is less convincing? Some recent reports, like Savigny et al. 2021 – Front Immunol. (https://doi.org/10.3389/fimmu.2020.588799), indicating a protective role of STING in IPF could be discussed. Overall, it could be worth mentioning that STING is also an immune suppressor (Sharma et al. 2015 – PNAS. https://doi.org/10.1073/pnas.1420217112), when describing conflicting or still unclear roles of STING in many conditions, as the author does when talk about bacterial modulation of STING.

Line 124 – The author says cGAS is not known to have a role in senescence beyond the cellular phenotype. This is not exact, as Dou et al. 2017 – Nature, (https://doi.org/10.1038/nature24050) showed cGAS is required for the SASP phenotype in vivo in mice.

Line 144 – Check the grammar for the sentence “in the increase the level…”

Line 171 – The author indicates only a single cGAMP transporter (correct name: SLC19A1), which is not the only one discovered and even not the most relevant for humans (based on current reports). SLC46A2 (Cordova et al. ACS Centr. Sci, 2021), LRCC8 (Zhou et al. Immunity, 2020) and P2X7R (Zhou et al. Immunity, 2020) could be also included in the text and/or in figure 1.

Line 188 – Check “DMXAASTING” spacing

Line 214 – There are studies that warn against global and unspecific STING stimulation in cancer treatment, suggesting that STING overstimulation might be bad as it could result in T cell death within tumors (i.e. Wu et al. 2020, Immunity - https://doi.org/10.1016/j.immuni.2020.06.009). This aspect could be worth mentioning as a condition to keep in mind while trying to design effective anticancer combinational therapies targeting STING.

Line 221 – SARS-Covid is wrong. Spell it as SARS-CoV2 to refer to the virus or Covid to refer to the disease.

Correct tumor side in tumor site at several positions

Author Response

I would like to thank this reviewer with the many excellent and detailed suggestions. I have tied to come up with amendments which I hope would satisfy most of the points. I really believe that the suggestions had helped largely improving the manuscript. Details of the changed in the revised manuscript are listed below.  I agree with the reviewer that I have been a little be "conservative" with the choice of the references, but the cGAS/STING has really exploded in the last years and it is nevertheless difficult to be fair, concise and comprehensive. I really valued the opinion of this reviewer, since the many references he suggests are complementing well the the body of the manuscript, extending some topics that due to size limitations could not be discussed in details. I have inserted a note on the many new references on lanes 219-221.

I agree with the review that work by Prabakaran et al on viral peptides is very interesting, but in my opinion and experience in drug development, I do not see these peptides as a chance for an entry into the clinics in the near future, which is what I have tried to focus on the review. That is why I would prefer not to discuss this topic, at least in the current context of this review.

Specific comments:

Line 22: change to "I", "we" has been used as "pluralis maestatis"

Lines 44 and 44: yes, agreed correct nomenclature is cyclic dinucleotide. Now corrected lines

Line 48 the paragraph has now been rewritten, as suggested by the reviewer. The reference to the review of Crow and Stetson has now been included.

Line 56: RVCL reference included in line 59

Line 68: Reference included in line 70

Line 70 (now line 72): done

Line 72-74: see now line 75-77 I have modified the initial statement, which might have been a little bit misleading. Vasculitis is clearly not only dependent on STING or type I IFN (many other possible pathways involved), but in SAVI patients pathology similar to vasculitis have been observed

Line 86: The reference by Huisjer has been included. This paper has just appeared while I was completing the review. I agree about the interesting suggestion about the hypersensitivity/exacerbated IFN-1 of monogenetic disease. I think the observation by Piperno et al is interesting. My feeling is nevertheless, that these observations might need to be discussed extensively in frame of other mechanisms that can control/modify cGAS/STING activity, which is beyond the scope of this review.

Line 93 (>now 96): TNF

Line 100: done

Line 112 (now line 113-116) I am personally not yet convinced that the evidence of cGAS/STING in lung fibrosis is strong. That it is what I have tried to summarize in the paragraph, which has now been modified. I have used the SAVI example to show the dichotomy of blocking/activating STING. I agree that there are many more examples that could have used here.

Lane 124 (line 127) the suggested reference by Dou et all has been included. I agree that this might indicated effects beyond cellular senescence. SAPS are considered biomarkers for senescence, but as clinical trails for ageing are very difficult to run, we still do not known how good SASP will translate an effect on human ageing.

Line 144 ( now line 146): "....result in the increase of the level result in the increase of the level..."

Line 171 (now 173): the references suggested by the reviewer for the other putative cGAMP transporters have been included. In the Fig.1, legend it has been included "(shown using the example of SLC19A1)" . SLC19A1 has been corrected in Fig. 1.

Line 188: (now 191) corrected

Line 214: as mentioned previously, a paragraph referring to additional references have been included (line 218-220). I agree that it will be important the potential considering the role of T cells, as mentioned in the paper by Wu et, I would need to include a section on the role of the cGAS/STING pathway in the T cells, which would require extending the length of this review. The challenges encountered in the clinical trials using cyclic dinucleotides (as STING agonist) were others. When we will have better compounds advancing in the clinic we will have the chance to also address the role of T cells in man.

Line 221 (now 227) corrected

Reviewer 3 Report

The manuscript did not contain sufficient new information to justify its publication: more comprehensive review articles of STING agonist and the related pathways already been published recently. For example: STING Agonists as Cancer Therapeutics, Cancers, 2021; Trial watch: STING agonists in cancer therapy, Oncoimmunology. 2020.

There are many typos in the manuscript. For example, line 96, what is “tonfa”? TNF alpha? Figure 1: the content of the blue rectangle is not complete and some parts are missing. Line 196, type I or type II IFN?

Author Response

I must disagree with this reviewer about the content of the manuscript. The purpose of the manuscript was not to give an overview of STING agonists in cancer. I agree with the reviewer that there excellent publications tackling this topic. In the revised version I have tried (also following the suggestions by other reviewers) to include many more of the references to STING agonists in cancer. Nevertheless the purpose (as also requested by the inviting authors) for this review was to give an overview of the potential of the STING agonists in cancer, but of STING low molecular weight compounds (with a special eye on STING antagonists) with potential to become in the near future useful therapeutics in diseases well beyond cancer and which would allow the explore the broad usefulness of this pathway in the clinics. 

Many typos (hopefully all) have been corrected.  

Round 2

Reviewer 3 Report

na